# Immunological Tolerance in Liver Transplant Recipients: Putative Involvement of Neuroendocrine-Immune Interactions

**DOI:** 10.3390/cells11152327

**Published:** 2022-07-29

**Authors:** Jaciara Fernanda Gomes Gama, Liana Monteiro da Fonseca Cardoso, Rodrigo da Cunha Bisaggio, Jussara Lagrota-Candido, Andrea Henriques-Pons, Luiz A. Alves

**Affiliations:** 1Laboratory of Cellular Communication, Oswaldo Cruz Institute, Oswaldo Cruz Foundation, Brazil Avenue, 4365-Manguinhos, Rio de Janeiro 21045-900, Brazil; bmd_gomes@yahoo.com.br (J.F.G.G.); lianamfc@gmail.com (L.M.d.F.C.); 2Laboratory of Immunopathology, Department of Immunobiology, Biology Institute, Federal Fluminense University (UFF), Gragoatá Bl-M Campus, Niterói 24210-200, Brazil; jlagrota@id.uff.br; 3Department of Biotechnology, Federal Institute of Rio de Janeiro (IFRJ), Maracanã, Rio de Janeiro 20270-021, Brazil; rbisaggio@gmail.com; 4Laboratory of Innovations in Therapies, Education, and Bioproducts, Oswaldo Cruz Institute, Oswaldo Cruz Foundation, Rio de Janeiro 21041-361, Brazil; andreah@ioc.fiocruz.br

**Keywords:** liver transplantation, regulatory microenvironment, neuroendocrine-immune interaction, adrenergic receptor, cholinergic receptor, immunological tolerance

## Abstract

The transplantation world changed significantly following the introduction of immunosuppressants, with millions of people saved. Several physicians have noted that liver recipients that do not take their medication for different reasons became tolerant regarding kidney, heart, and lung transplantations at higher frequencies. Most studies have attempted to explain this phenomenon through unique immunological mechanisms and the fact that the hepatic environment is continuously exposed to high levels of pathogen-associated molecular patterns (PAMPs) or non-pathogenic microorganism-associated molecular patterns (MAMPs) from commensal flora. These components are highly inflammatory in the periphery but tolerated in the liver as part of the normal components that arrive via the hepatic portal vein. These immunological mechanisms are discussed herein based on current evidence, although we hypothesize the participation of neuroendocrine-immune pathways, which have played a relevant role in autoimmune diseases. Cells found in the liver present receptors for several cytokines, hormones, peptides, and neurotransmitters that would allow for system crosstalk. Furthermore, the liver is innervated by the autonomic system and may, thus, be influenced by the parasympathetic and sympathetic systems. This review therefore seeks to discuss classical immunological hepatic tolerance mechanisms and hypothesizes the possible participation of the neuroendocrine-immune system based on the current literature.

## 1. Introduction

### The Immune System in Neuroendocrine-Immune Crosstalk: Early Evidence

The development of the immune system, like that of other homeostatic systems, has evolved over millions of years after the appearance of the first metazoans (animals). In mammals, this system allows for integrity maintenance, responding to non-self components and altered self or damage-associated molecules [1]. Therefore, immune surveillance recognizes dangers, including cancer cells, foreign substances, infectious pathogens, and even self-components.

During metazoan evolution, the immune system developed alongside several cellular communication systems, such as the nervous system, with action potentials and superfast messenger transmission (<5 ms-electrical synapse) [2]; the endocrine system, which provides hormones that act over long distances; and paracrine or synaptic intercellular communication, which acts over short distances. However, historically, immunology and physiology have always been studied separately [3,4]. The study of the immune system evolved in association with the study of bacteriology and microbiology, later becoming an independent discipline. The same was observed in physiology, which was associated with growing knowledge of anatomy and medicine. These separate developments restricted the contact between immunologists and physiologists. Thus, for many years in the past, it was impossible to imagine that the immune system could influence brain activity and the endocrine system or vice versa.

One of the first clues of neuroendocrine-immune loops was obtained by Hadden in 1970 [5] for evidence on the expression of adrenergic receptors in human peripheral lymphocytes, suggesting a link between the sympathetic nervous system and the immune response. Moreover, as established by one of the pioneers in this area, neuroendocrine-immune interactions first gained recognition at an immunology conference held in Canada in 1986. Thus, Basedovsky and coworkers revealed feedback regulation between interleukin (IL)-1 and glucocorticoids in an inflammatory microenvironment [6,7]. Then, Ader and coworkers demonstrated the influence of “psychological” mechanisms on the immune system using behavioral experiments [8].

Considering these data and the fact that the β2 adrenergic receptor, as well as some cholinergic receptors, are widely expressed in T lymphocytes, it is possible that neuroendocrine-immune crosstalk plays a role in liver transplantation (LT) tolerance. The expression of these receptors was evidenced mainly through the effects of agonists and antagonists [9,10].

## 2. Background on Liver Transplantation

The discovery and implementation of calcineurin inhibitors, in addition to cyclosporine, in the early 1980s, followed by tacrolimus in the 1990s, significantly reduced acute cellular rejection (ACR) and dramatically improved the one-year patient survival rate [11,12]. Other agents, such as purine analogs and mammalian target of rapamycin (mTOR) inhibitors, have been developed, further lowering ACR rates and calcineurin inhibitor-related toxicity. Additionally, patient survival has continued to improve over the last decade, although ACR remains a challenge, mainly when occurring late after LT [13,14]. The induction of graft immune tolerance is a complex process essential for minimizing organ or tissue/cell rejection after transplantation. LT is the only effective treatment for end-stage liver diseases. Following transplantation, the use of immunosuppressants (ISs) is an approach used to prevent acute immune rejection and modulate immune tolerance to the transplanted organ. In recent years, there has been a substantial improvement in the patient survival rate following LT, mainly due to advances in ISs therapy. Despite improved pharmacological strategies to prevent acute transplant rejection, ISs cause countless adverse effects, leading to an increased malignancy and opportunistic infections, and promoting or enhancing cardiac, renal, and metabolic pathological conditions. In addition, they increase health care costs, require drug monitoring, and usually increase long-term morbidity and mortality [15]. Over the years, many strategies have been implemented to reduce these effects, such as gradually reducing IS administration and applying cell-based therapies to induce immune tolerance to grafts, which can, in turn, minimize liver rejection [16,17,18]. It is important to highlight that the interplay between the immune system and neuroendocrine responses has been increasingly associated with immune tolerance in various conditions; however, it remains unclear after LT. Currently, there is a consensus that the human psychological status can affect immunological responses. However, we do not yet understand how to control these mechanisms, either psychologically or pharmacologically. Thus, understanding these pathways may be helpful for improving LT therapies and the patient quality of life and survival. Thus, in this review, we discuss neuroendocrine-immune interactions and their possible applications to prolong liver tolerance after transplantation, since the evidence of this axis in the context, as well as the tolerance after LT, is still poorly understood.

## 3. Current Ideas of Liver Tolerance Mechanisms after Transplantation

Currently, ISs, including calcineurin inhibitors (cyclosporin and tacrolimus) and corticosteroids that target the activation, expansion, and cytotoxicity of the recipient’s T lymphocytes, have created improvements in transplant surgeries since the 1970s and have reduced the acute rejection rate to less than 15% of transplants, but long-term ISs use is associated with increased risks of infections and malignancies [19,20,21]. Preclinical experiments in animal models were essential for understanding the mechanisms of rejection or tolerance of LT. It was observed that hepatic allografts could be accepted by MHC-mismatched individuals without IS treatment for a short time. Moreover, the prior transplantation of liver fragments induced immunological tolerance to secondarily transplanted solid organs from the same donor. In contrast, when solid organs from another donor were transplanted, they were rejected [22,23,24]. Therefore, the liver appears to provide some level of immunological protection for other organs in cases of combined organ transplantation (liver-kidney or liver-intestine), with the second solid organ being less adversely impacted by donor-specific antibodies [25,26,27]. Both cellular and humoral alloimmune responses contribute to rejection. It is also important to know that LT itself is capable of inducing inflammatory pathways as the hepatic ischemia-reperfusion injury [28]. The liver microenvironment passes by waves of the proinflammatory and anti-inflammatory response that remains throughout life, and this regenerative profile, as well as the cytokines subtypes secreted, is closely related to the restoration of liver function and post-LT clinical outcomes [29].

Clinically, LT recipients require the lowest amounts of ISs and exhibit the lowest incidence of chronic immune-mediated injury [30] compared to other organ recipients, and have a propensity toward tolerance over alloreactivity. Interestingly, approximately 38% of adults (>18 years old) and 44% of children (<18 years old) among all patients who undergo LT can discontinue IS therapy in the longer term, as shown in Table 1.

## 4. Immune Cells and Liver Transplant Immune Tolerance

Clinical trials have provided clues to clarify the roles of immune cells in LT, with the possibility of developing cell-based therapies. As discussed earlier, tapering to complete withdrawal may be a viable strategy. In addition, cell therapy would contribute to an increased tolerance after LT, such as in clinical trials that use Treg cell therapy [41]. Thus, since Treg cells are the main cells involved in tolerance after LT and their involvement in LT tolerance was successfully observed in humans [18,42,43], most clinical studies have targeted these cells (Table 2). Thus, immune cells have been studied with the aim of inducing tolerance mechanisms, as discussed below.

### 4.1. Natural Killer T Cells

Experimental studies in rats showed that natural killer T (NKT) cells, an abundant cell present in liver, could play an important role in liver transplant tolerance by promoting a Th-1 shift to Th-2. Allograft survival was significantly increased in mice inoculated with α-galactosylceramide, a synthetic glycolipid that induces NKT cells activation, increasing IL-10 and decreasing IFN-γ levels [44].

### 4.2. Natural Killer Cells

Donor liver natural killer (NK) cells may influence the allograft acceptance. They act as “passenger leukocytes”, circulating in the recipient’s body and inducing tolerance [52]. In this sense, the liver from IL-4-treated donor rats is less rejected. IL-4 induced an extensive inflammatory infiltrate in the donor liver consisting of alternately activated macrophages and IDO (indoleamine dioxygenase)-expressing NK cells with potentially immunosuppressive activity and associated with migration to a recipient spleen [45].

### 4.3. Dendritic Cells

In animal models, the intraperitoneal or venous injection of immature DCs overexpressing IL-10 or Fas-L was able to prevent liver damage after LT, probably by inducing Treg cells [47,48,49]. In post-LT mice, DCs showed an increase in PDL-1 expression (Figure 1), which contributed to a decrease in the T cell response through the acquisition of MHC molecules via the “cross-addressing” of donor cells by host cells [50,53]. In humans, following DC infusion in patients who underwent LT (clinical trials: NCT03164265 [50] and NCT04208619), alloreactive cells against donor cells were suppressed with the maintenance of the regulatory environment [50].

### 4.4. Regulatory T Cells

Treg cells play essential roles in the mechanisms of immunological tolerance in LT in experimental allograft model animals, such as rats [54] and mice [55], as well as in humans [18,56]. In addition to acting through the inhibitory receptor CTLA-4, also known as CD152, Treg cells exert their suppressive functions through the expression of IL-10 and TGF-β (Figure 1). It is important to highlight that CD4^+^ Treg cells compose approximately 10% of peripheral lymphocytes in humans, and their phenotype is CD25^high^FOXP3^+^ [57].

Teratani and colleagues [58] described the liver-brain-gut interaction in the Treg induction on colitis model as contributing to immune tolerance in the peritoneal cavity through the hepatic vagal sensory afferent nerves. It indirectly induces the sensory inputs to the brainstem on the nucleus tractus solitarius, and then to vagal parasympathetic nerves and enteric neurons that lead the Treg niche maintenance. In line with this idea, they also demonstrated that this alteration could reduce the Treg and the tolerance effects in the inflammatory bowel microenvironment. Although there are no studies demonstrating the activation of Treg cells in tolerance after LT through adrenergic receptors, it is possible to consider the importance of the neuroendocrine-immune axis in immunological tolerance in diseases of the peritoneal cavity via the activation of β-adrenergic receptors on Treg cells in colitis [59], inflammatory bowel disease [60], or experimental autoimmune encephalomyelitis [61]. These findings could suggest the participation in the activation of the regulatory population via β-adrenergic receptors on spontaneous natural tolerance after LT. In addition, in previous clinical trials, such as the infusion of donor alloantigen-reactive Treg cells (clinical trials: NCT02474199, NCT02166177, and NCT03577431) or autologous Treg cells (NCT02166171), the results showed that this procedure is safe and induces tolerogenic microenvironment. However, some results have not yet been disclosed. In addition, a clinical trial using regulatory DCs showed that these cells induced a hyperresponsiveness in effector T lymphocytes and an enhanced Treg cell function [62]. In spontaneously tolerant animal models of hepatic transplantation, CD4^+^CD25^+^ T cells expressing FOXP3 are significantly increased in relation to acute rejection models, indicating that Treg cells might be involved in the induction of spontaneous immune tolerance [51].

Understanding the participation of the immune cells mentioned above, in the context of LT and how they could be related to neuroendocrine interactions, is important to suggest modulation via this axis, considering that the liver has a tolerogenic microenvironment. In addition, they have receptors and are influenced by molecules from the nervous system. These interactions will be briefly described below.

## 5. The Hepatic Tolerogenic Microenvironment and the Interplay with the Nervous System

As mentioned previously, many immune cells are part of the liver microenvironment (Figure 2), such as NK and NKT cells, which compose more than 50% of hepatic lymphocytes, followed by conventional circulating and intrahepatic CD8^+^, CD4^+^, and γδ T lymphocytes. Kupffer cells (KCs) compose approximately 20–35% of nonparenchymal liver cells, and different populations of DCs, such as plasmacytoid DCs (pDCs), and lymphocytes are found in the liver and have distinct phenotypes depending on their origin [63,64,65]. In addition, these immune cells from the liver microenvironment have surveillance features, as they widely express pattern recognition receptors (PRRs), such as scavenger receptors, carbohydrate receptors (lectins), TLRs, and cytoplasmic receptors, that are capable of responding to blood and gut antigens. There is plenty of evidence that antigens that enter the liver may lead to a natural bias toward tolerance through the production of anti-inflammatory mediators and expression of inhibitory cell surface ligands [66,67].

### 5.1. Kupffer Cells

Hepatic tolerance is maintained with the participation of different hepatic cell populations [68], such as KCs (Figure 2) [69]. KCs compose approximately 20–35% of nonparenchymal liver cells.

KCs are anatomically located in the lumen of the hepatic sinusoids, a network of fenestrated blood vessels lined by LSECs [65]. KCs express major histocompatibility complex (MHC)-I and MHC-II and the costimulatory molecules B7.1, B7.2, and CD40, although at lower levels than HDCs. Under steady-state conditions, these cells secrete transforming growth factor-beta (TGF-β), prostaglandin E2 (PGE2), and IL-10 [70]. In addition, they express Fas-L and programmed cell death-ligand (PD-L1), a potent inhibitor of immune responses that also downregulates T lymphocyte function after binding to programmed cell death protein-1 (PD1) on the T cell membrane [71]. This repertoire of secreted and surface molecules leads to the differentiation of hepatic regulatory T (Treg) cells [72], a population highly represented in the liver. In vivo, KCs induce apoptosis in neutrophils and other polymorphonuclear cells (PMNCs) through the Fas/Fas-L pathway [73]. The engagement of phosphatidylserine (PS) exposed by apoptotic cells with the PS receptor on KCs has also been shown to lead to the secretion of more TGF-β, IL-10, and PGE2 [74]. Moreover, this interaction reduces the production of proinflammatory cytokines by KCs under inflammatory conditions [75] and contributes to liver tolerance maintenance.

Sympathetic nerve fibers can modulate hepatic inflammation by adrenergic receptors expressed by KCs through the activation of α1-ARs and α2A-Ars, inducing an increased inflammatory cytokine production, whereas the activation of β2-Ars on KCs decreases the production of these mediators. In this sense, sympathetic denervation or blocking α1-ARs in KCs reduces the production of cytokines, such as IL-6 and TGF-β, and reduces the development of HCC (hepatocellular carcinoma) [76]. In this neuroendocrine-immune crosstalk context, it was demonstrated that the N-methyl-D-aspartate (NMDA) receptor is expressed by KCs and that its activation on primary mouse and human cells has an anti-inflammatory activity, limiting injury in acute hepatitis in vivo [77]. This effect was observed to be due to the downregulation of NOD-like receptor family, pyrin domain containing 3 (NLRP3) and procaspase-1, leading to the downregulation of inflammasome activation via a β-arrestin-2, nuclear factor kappa B (NF-kB)-dependent pathway, and not via Ca^2+^ mobilization [77]. These observations highlight important roles for the complex interactions between the immune system and neural response in inflammatory and anti-inflammatory pathways as possible keys to new therapeutic approaches.

### 5.2. Hepatic Dendritic Cells

HDCs are a very heterogeneous population in the liver, performing multiple functions under normal conditions or after transplantation [78]. In the liver, some cytokines, such as FMS-like tyrosine kinase 3 ligand (Flt3L) and GM-CSF, can recruit conventional DCs (cDCs) originating from bone marrow progenitors [78]. However, in the case of monocyte differentiation into HDCs in the liver, the intrahepatic environment induces the differentiation of an HDC subset that leads to Th2 responses [79]. Some anti-inflammatory and immunosuppressive drugs can affect DC recruitment to the liver and HDC maturation and function; these drugs include aspirin, corticosteroids, calcineurin inhibitors, and rapamycin [78,80]. HDC subsets include pDCs and cDCs (also known as myeloid dendritic cells), which are subdivided into cDC1s (CD8^+^ lymphoid) and cDC2s (CD11b^+^) [81].

HDCs, in general, secrete lower levels of IFN-γ than extrahepatic DCs and more IL-10 than IL-12, favoring Th2 responses [82,83]. Under normal conditions, hepatic pDCs are relatively immature antigen-presenting cells (APCs) with a lower endocytic capacity and lower expression of MHC-II [84] and costimulatory molecules, such as CD40, B7.1, and B7.2 [85,86]. The other hepatic DC subpopulations express higher levels of these molecules. However, HDCs may express high levels of PD-L1, TGF-β, PGE2, and other immune-inhibitory molecules, maintaining hepatic immune tolerance [87].

The neuroendocrine-immune interactions in the peritoneal cavity and the influence of neurotransmitters and neuropeptides on DCs are still poorly understood. The anatomical proximity of HDCs to the nerve peripheral plexus suggests a response mediated via neuroendocrine-immune interactions, which possibly have a relatively great influence on NPY and NE released from hepatic sympathetic nerves [88]. Unfortunately, there is little and contradictory evidence demonstrating the influence of neuropeptides or neurotransmitters on DCs involved in the activation of pro- and anti-inflammatory pathways, probably due to the heterogeneity of the models studied and their induced immune responses.

The release of serotonin (5-HT), a neurotransmitter monoamine related to the pathophysiological mechanisms of inflammatory disease in the gastrointestinal tract, increases inflammation in the context of intestinal inflammation, probably via the serotonin receptors 5-HTR1B and 5-HTR2A on DCs, increasing proinflammatory cytokines in activated B cells via NF-kB [89,90]. In contrast, the deletion, mainly in DCs, of 5-HTR, or even blockade with SB-269970, a 5-HTR antagonist, was shown to increase the severity of inflammation in colitis models, increasing the production of proinflammatory cytokines (e.g., IL-1β, IL-6, and TNF-α) [90,91], showing the possible close relationship between the neuronal mediators and the modulation of the immune system in the liver by DCs. Thus, these data suggest that the role of DCs via neuroendocrine-immune interactions may be related to the modulation of the immune response in the liver, but further studies are needed to clarify how this occurs.

### 5.3. Hepatic Stellate Cells

HSCs (Figure 2) are best known for their capacity to store vitamin A and retinyl esters [92], but they are also considered as functional APCs in the liver. At least in culture, HSCs express members of the HLA family (HLA-I and HLA-II), lipid-presenting molecules (CD1b and CD1c), and accessory molecules involved in T-lymphocyte activation (CD40 and B7.1) [93]. Moreover, these characteristics were shown to be increased after incubation with proinflammatory cytokines such as IL-1β and IFN-γ, and these cells could efficiently present antigens to CD1d-, MHC-I-, and MHC-II-restricted T lymphocytes [94]. In contrast, it was also shown that HSCs could inhibit T cell responses via PD-L1-mediated apoptosis. Moreover, HSCs alone do not seem to present antigens to naïve CD4^+^ T lymphocytes, but, in the presence of HDCs and TGF-β, they preferentially induce FOXP3^+^ Treg cells [95]. Finally, HSCs seem to be essential in generating fibrosis caused by multiple etiologies [96], a feature that could be observed after LT.

Roskmans and coworkers hypothesized that HSCs and hepatic progenitor cells form a neuroendocrine compartment in the liver, expressing neuronal proteins, such as neural cell adhesion molecules, neurotrophin, and their receptors [97]. In addition, HSCs express the serotonin receptors 5-HT2A and 5-HT2B; these cells also take up and release serotonin through the 5-HT receptor [98]. These receptors contribute to hepatic fibrosis, HSC proliferation, gene transcription, and apoptosis [90,99]. Human HSCs also respond to NE, as cellular exposure to this neurotransmitter triggers pro-inflammatory responses with the secretion of inflammatory chemokines, such as RANTES and IL-8, and calcium spikes, which were partially attenuated with the administration of a nonspecific beta-blocker (propranolol) [100]. However, this activation is selectively suppressed by α1-B and α1-D adrenoceptor antagonists [101]. Thus, HSCs can communicate with the nervous system, modulating the immune response.

NK cells [102], NKT cells [103], and LSECs [104,105] also play roles in the maintenance of the tolerogenic status of the liver and its intrinsic pathways involved in triggering controlled inflammatory responses.

Most cell populations found in the liver and their biochemical regulatory pathways lead to Th2-biased immune responses and the differentiation of Treg cells that play key roles in the liver regulatory pattern. These cells exert this function through IL-4, IL-10, and TGF-β production, as well as indoleamine 2,3-dioxygenase (IDO1), PD-L1, and cytotoxic T-lymphocyte-associated protein 4 (CTLA-4) activity (Figure 2) [106]. In addition, Treg cells can suppress the activation of alloreactive conventional CD4^+^ and CD8^+^ T cells, as observed in a murine model of allogeneic transplantation [107]. These intercellular interactions certainly contribute to the tolerogenic response after LT and to minimizing IS administration and improving immune tolerance. Thus, it is important to understand the liver innervation and the mechanisms related to modulation via neuropeptides and neurohormones.

## 6. The Autonomic Nervous System in the Liver

The liver is innervated by afferent and efferent nerve fibers of either sympathetic or parasympathetic origin [108] that employ multiple neurotransmitters that are recognized by most, if not all, hepatic cell populations. The possible endocrine role of these neurotransmitters in the metabolic pathway may integrate the liver, brain, and periphery.

In the case of successful human LT, implying the complete autonomic liver denervation and ablation of the neuronal brain-liver connection, accumulating evidence indicates changes in liver metabolism. In 1848, the French physiologist Claude Bernard observed a decrease in hepatic glucose output after peripheral vagotomy, the earliest known study on hepatic autonomic innervation and glucose metabolism [109]. Many years later, in 1969, Niijima observed that, upon glucose infusions into the portal vein in guinea pigs, a change in vagal activity was recorded [110]. These pioneering works showed the interplay between the liver and the nervous system, but much still remains to be clarified, including its role in tolerance.

The distribution of sympathetic and parasympathetic nerves in the liver is highly variable among species. In humans, these nerves are observed surrounding the hepatic artery, portal vein, and bile ducts, extending into the hepatic lobules and reaching liver parenchymal cells [111]. Through autonomic nervous system (ANS) fibers, a bidirectional connection is mediated between the liver and the central nervous system (CNS), a characteristic that can affect liver metabolic and immunological responses. The sympathetic splanchnic nerves that innervate the liver originate from neurons in the celiac and superior mesenteric ganglia (Figure 3). The parasympathetic nerves mainly originate from preganglionic neurons in the dorsal motor nucleus of the vagus, which is located in the dorsal brainstem (Figure 3) [112]. Sympathetic and parasympathetic efferent nerves in the liver contain aminergic epinephrine and norepinephrine (NE), in addition to cholinergic neurotransmitters and peptidergic components, such as neuropeptide Y (NPY) (Figure 3) [109].

The efferent hepatic nerves act in the regulation of multiple hepatic physiological functions, including the contractility of the sinusoids, as observed in dogs [113]. It was demonstrated that nerve endings containing aminergic, peptidergic, and cholinergic neurotransmitters terminate near HSCs in the vascular walls, affecting the vascular caliber [114]. The sympathetic release of epinephrine leads to the contraction of the sinusoids, whereas the parasympathetic release of acetylcholine (ACh) and vasoactive intestinal peptide (VIP) induces vascular relaxation [115]. Regarding liver regeneration, it was observed that, after partial hepatectomy, liver regeneration was severely impaired by vagotomy in rats [116]. Moreover, adrenergic signaling also stimulates liver regeneration [117] through hepatocyte growth factor [113]. In vitro, TGF-β was shown to be a potent inhibitor of epithelial growth factor (EGF)-induced DNA synthesis in primary rat hepatocytes, whereas NE was shown to counter-modulate this inhibition [118]. The consequences of nerve ablation on the regenerative capacity of hepatocytes are particularly important for understanding liver engraftment and postsurgical IS regimens, which remain poorly investigated.

Few published papers have studied liver regeneration and re-enervation after partial or total hepatectomy, a very important aspect in determining the postoperative LT outcome. Using rats that underwent partially hepatectomy, it was observed that, after subdiaphragmatic vagotomy, hepatic DNA synthesis and thymidine kinase activity were delayed [119]. Moreover, subdiaphragmatic vagotomy caused a considerably greater loss of food intake and body weight, whereas hepatic vagotomy led to no alterations in either parameter. Similar results were obtained by another group, with the increase in DNA synthesis after partial hepatectomy being markedly suppressed and delayed by subdiaphragmatic vagotomy [116]. All of these results illustrate the complexity of the postoperative and general management of chronic liver disease patients and the necessity to discuss these issues further.

## 7. Neuroendocrine-Immune Modulation: A Complex Process toward Tolerance

Cells of the immune system have been described since the 1980s as having the capacity to release these neuronal molecules [120], which may regulate tolerance processes. Guereschi and coworkers showed that Treg cells express the β2-adrenergic receptor (Table 3) and that its activation increases the suppressive activity in a protein kinase A (PKA)-dependent pathway, upregulating Treg cell differentiation and cyclic adenosine monophosphate release [121]. Thus, neurohormones and neurotransmitters are released by the nervous system and can modulate immunological functions.

The interactions between immunoregulatory cells and the nervous system may be key factors in understanding immunological tolerance mechanisms involved in transplantation and autoimmune diseases. Furthermore, psycho-emotional illnesses and bio-psychosocial disorders, such as maternal-fetal deprivation in the early years, play important roles in inducing the intestinal inflammatory response and in Crohn’s disease rates, suggesting a possible inflammatory modulation loop via brain-gut interactions [133,134]. Patients with end-stage cirrhotic liver disease and hepatitis and post-transplant patients have a high incidence of depression, a condition that may be controlled by a reduction in hepatic encephalopathy through the use of antidepressants [135].

Surgical vagotomy was able to reduce dysbiosis in mice with CCl4-induced cirrhosis, but these animals showed increased levels of brain-derived neurotrophic factor, a key protein involved in the pathogenesis of cirrhosis and associated complications, and inflammatory cytokines such as IL-1β and MCP-1, as well as increased liver steatosis [136]. In keeping with this idea, a model of dextran sulfate sodium-induced colitis showed that vagotomy contributed to disease worsening and increased inflammation, with increasing NF-kB levels and decreasing Treg cell numbers levels [137]. In contrast, Wirth and colleagues demonstrated that lpr-lpr mice, which have autoimmune lymphomyeloproliferative disease, and C57BL/6 mice with 6-OHDA-induced chemical sympathectomy had lower levels of NE and increased levels of Treg cells colocalized in the splenic nerves, suggesting that blocking the neurosympathetic pathway could contribute to disease improvement [138]. These controversial responses may be related to the complexity of and differences in the neural and immune responses in these models.

It is important to highlight that vagal nerve stimulation (VNS) has contributed to clarifying this evidence and has become a promising nondrug treatment in bioelectronic medicine. For example, VNS was able to ameliorate pathogenesis and decrease the levels of markers related to disease progression in murine models of collagen-induced arthritis [139] and trauma-hemorrhagic shock [140]. These effects occur mainly through cholinergic anti-inflammatory pathways (CAIPs), since cholinergic agonists inhibit the release of TNF-α and other cytokines by macrophages via interaction with α7nAChR (Figure 1) [141]. This increases Treg cells and NE release or increases the Treg/Th17 cell ratio and decreases TNF-α, respectively [139,140]. Furthermore, a pilot study on Crohn’s disease patients showed that chronic VNS was able to alleviate disease severity and decrease C-reactive proteins through the activation of CAIP [142]. These findings indicate the important role of immune modulation mediated through neuroendocrine-immune communication. Once better understood, exploitation of this communication will contribute to the advancement of new therapeutic methodologies, such as VNS, and the improvement of the above-described diseases.

## 8. The Neuroendocrine-Immune Crosstalk in the Liver May Possibly Influence Tolerance after LT

Neurotransmitters are synthesized in neurons and present in the presynaptic terminal portion of these cells. In addition, these chemicals must exert a specific action on postsynaptic neurons or target cells in effector organs, exogenous components should mimic their action, and cellular mechanisms should remove them from the intersynaptic cleft [143]. Neurochemicals then act on cells that express the corresponding specific receptors, including postsynaptic neurons, T lymphocytes, NK cells, and many other liver-resident cell types. Thus, the premise for neuroendocrine-immune tolerogenic action in the liver is the exposure of hepatic cells to neurotransmitters and other neurochemicals, as intercellular communication in the nervous system is based on these messengers, as previously mentioned.

In the liver, dopamine plays an essential role in suppressing autoimmune hepatitis [144]. A previous study showed that the depletion of dopaminergic neurons led to hepatic invariant NK (iNKT) cell activation and augmented concanavalin A-induced liver injury. This suppressive effect of dopamine on iNKT cells was mediated by the D1-like receptor-PKA pathway and gut microbiota, as antibiotic administration reduced dopamine synthesis in the intestines and exacerbated liver damage. These results suggest not only that intrahepatically produced neurochemicals may affect liver cell populations and functions but also that dopamine acts on G-protein-coupled receptors (D1–D5), and different subtypes have been described in human and mouse lymphocytes [145] and have been shown to affect lymphocyte proliferation, activation, fibronectin adhesion, chemotaxis, and function [146].

The neurotransmitter ACh has also been described as having suppressive actions on the immune system, and CAIPs have been observed in the liver [147]. The release of ACh following vagal efferent fiber activation leads to the inhibition of inflammatory cytokines through α7nAChRs, which are located on the surface of KCs and other cells in the liver. This communication seems to be mediated by the prior release of NE, which interacts with β2-adrenergic receptors (Table 3) and causes the release of ACh by T cells. ACh then interacts with α7nAChRs on macrophages and suppresses proinflammatory cytokine release and inflammation [148]. It was also demonstrated that PNU-282987, a selective α7nAChR agonist, protects the liver from ischemia-reperfusion injury by inhibiting NF-kB activation in mice [149]. Based on this, the liver is innervated by vagus branches, and inflammatory responses may be modulated by the activation of vagal efferent fibers.

The neurotransmitter gamma-aminobutyric acid (GABA) is synthesized by the decarboxylation of glutamate by the enzyme glutamine acid decarboxylase (GAD), which has two isoforms, GAD65 and GAD67 [150]. In addition to cells from the nervous system, T lymphocytes and macrophages secrete GABA [151], an inhibitory neurotransmitter. By PCR, two GABA receptor subunit types (β3 and ε) were detected in human hepatocytes. Moreover, increased GABAergic activity was associated with a reduced hepatocyte proliferation and attenuation of hepatic regeneration after partial hepatectomy [152]. Conversely, decreased GABAergic activity was associated with enhanced hepatic regeneration after ethanol exposure or toxin-induced acute or chronic liver disease [153,154,155].

It was reported that platelet-derived serotonin supported viral persistence in the liver and aggravated virus-induced immunopathology after experimental infection with noncytopathic lymphocytic choriomeningitis virus (LCMV) [156]. This suggests that serotonin may also be involved in controlling the hepatic inflammatory response. In this case, platelets accumulated in the liver and severely reduced the sinusoidal microcirculation, delaying LCMV elimination and increasing liver damage. The serotonin treatment of infected mice delayed CD8^+^ T lymphocyte migration to the liver and aggravated immunopathological hepatitis [156]. These results were confirmed in infected serotonin-deficient mice and showed the direct and indirect effects of neurotransmitters on the hepatic pathophysiology. Moreover, serotonin modulation in the gut-liver neural interaction ameliorated fatty and fibrotic changes in nonalcoholic fatty liver disease (NAFLD) [157].

It has also been reported that stress-induced liver injury can be stimulated by NE, sympathetic nerve activation, and stress hormones [158]. Circulating and locally secreted hormones also play an immunoregulatory role in the liver. For example, during pregnancy, when maternal FOXP3^+^ Treg cells are expanded, the immune response shifts toward a Th2-dominant arm, and other immune-endocrine adaptations occur. During the gestational period, autoimmune hepatitis remission occurs, which results in a postpartum flare of this pathological condition [159]. These results indicate the great impact in the neurological vertices by hepatic metabolic and immunological functions and show that much remains to be understood.

Considering that much of the evidence on the interaction between the ANS and liver diseases is focused on portal pressure, the imbalance in the heart rate, and characteristics of encephalopathy, little is known about the role of the ANS in the immune-mediated hepatic response, especially after LT. Thus, given the extensive hepatic innervation, as well as the known roles of neurotransmitters and neuropeptides in immune cells, as described above, it is possible that there is a close involvement of neuronal stimuli in the mechanism of hepatic tolerance after LT. This information could contribute to new therapeutic methodologies capable of improving patient survival and the quality of life after LT.

## 9. Conclusions

The hepatic microenvironment exerts an important influence on the mechanisms involved in many conditions that affect the liver (e.g., autoimmune and viral hepatitis, cytomegalovirus infection, NASH, cirrhosis, and other conditions), and the mental state may also be related to this imbalance. Thus, the resident and transient cells present in the liver, such as HSCs, LSECs, HDCs, and intrahepatic and circulating lymphocytes, contribute to the immunosurveillance and immune tolerance modulation that occur in the liver. Although there are no definitive studies showing the neuroendocrine-immune interaction interplay as a mechanism of tolerance after LT, understanding how neurotransmitters and neurohormones influence the liver microenvironment may help to develop an approach to induce immunological regulation. The neuronal, endocrine, and immunological responses contribute to the highly complex intercellular interactions that occur in the liver, which may be explored as a potential new therapeutic perspective to improve the acceptance of transplanted livers and extend the quality of life and expectancy of chronic hepatitis patients. Thus, although the neuroendocrine-immune interactions occurring after LT are unknown, they may play key roles in modulating the mechanisms involved in LT tolerance.

Furthermore, considering bioelectronic medicine, nondrug intervention through VNS may be a strategy, with little invasiveness required to induce neuroendocrine-immune communication for tolerogenic immunoregulation in the liver. These aspects may help to reduce or discontinue IS administration and prolong the life expectancy in transplanted patients.

## Figures and Tables

**Figure 1 cells-11-02327-f001:**
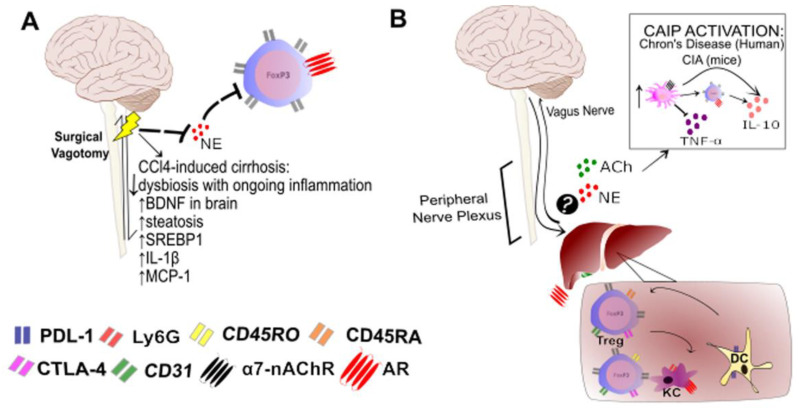
Possible influence of the nervous system on the liver immune system after liver transplantation. (**A**) Among the interactions of neurotransmitters in liver injury by CCl_4_-induced cirrhosis, a decrease in norepinephrine (NE) is observed after sympathectomy, followed by a decrease in regulatory T cell (Treg) and increases in inflammatory cytokines, such as interleukin (IL)-1β and monocyte chemoattractant protein-1 (MCP-1), in addition to increases in hepatic steatosis and brain inflammation markers. (**B**) Collectively, evidence for neuroendocrine-immune interactions in the liver, mainly collected in clinical trials, shows that, after liver transplantation (LT), the Treg cell pool is heterogeneous and may present phenotypes indicative of different origins. Thus, these cells may express CD45RO or CD45RA, in addition to presenting CD31 on the cell surface. In addition, dendritic cells (DCs) contribute to the regulatory microenvironment, leading to a tolerogenic response after LT. In the peritoneal microenvironment, it is possible that these cells are modulated by β-adrenergic receptors in Kupffer cells (KC) via NE, as observed in other conditions associated with the peritoneal microenvironment or acetylcholine (ACh) release induced by cholinergic anti-inflammatory pathway (CAIP) activation. These changes are able to induce the anti-inflammatory macrophage profile via Treg cells and thus modulate the inflammatory response. BNDF: brain-derived neurotrophic factor; SREBP1: sterol regulatory element-binding protein-1; PDL-1: programmed death-ligand 1; CTLA-1: cytotoxic T-lymphocyte-associated antigen 4; α7nAChR: alpha7-nicotinic acetylcholine receptor; AR: adrenergic receptor.

**Figure 2 cells-11-02327-f002:**
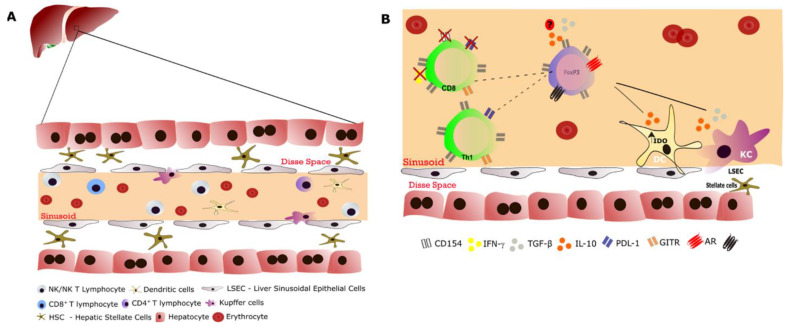
Liver cells in immunity: hepatic cells in homeostasis and the regulatory microenvironment in the immunosuppressive response after LT. (**A**) The liver is an organ composed of parenchymal (hepatocyte) and nonparenchymal cells that exert different functions. During homeostasis, nonparenchymal cells such as Kupffer cells, dendritic cells, NK cells, NKT cells, and HSCs constitute the immunological liver microenvironment, responding to most gut-derived antigens. In addition to these resident cells, there are transient lymphocytes in the sinusoidal space. In addition, hepatocytes can play an immunological role in the context of innate protein release. (**B**) Regulatory T (Treg) cells play an important role in the mechanisms of allogeneic response suppression after LT. Kupffer cells (KCs) and dendritic cells (DCs) have important roles in this regulatory microenvironment mediated through IL-10 production and TGF-β release. These molecules contribute to the induction of Treg cells, which probably act directly to decrease the response via effects on CD4^+^ (Th1 cells) and CD8^+^ T cells, minimizing the rejection process. In CD8^+^ T cells, there is a decrease in CD154, an important protein expressed on the surface of activated cells in humans. Furthermore, in a mouse model, it was observed that cytotoxic T-lymphocyte-associated antigen 4 (CTLA-4) induction in Treg cells promoted an increase in glucocorticoid-induced TNFR-related protein (GITR) on CD8^+^ and CD4^+^ T cells, decreasing the rejection responses of these T cells through the induction of apoptosis in these cells, mainly via increased expression of programmed death-ligand1 (PDL-1). Thus, the production of inflammatory cytokines, such as IFN-γ, was decreased. LSEC: liver sinusoidal endothelial cells; IDO: indoleamine 2,3-dioxygenase.

**Figure 3 cells-11-02327-f003:**
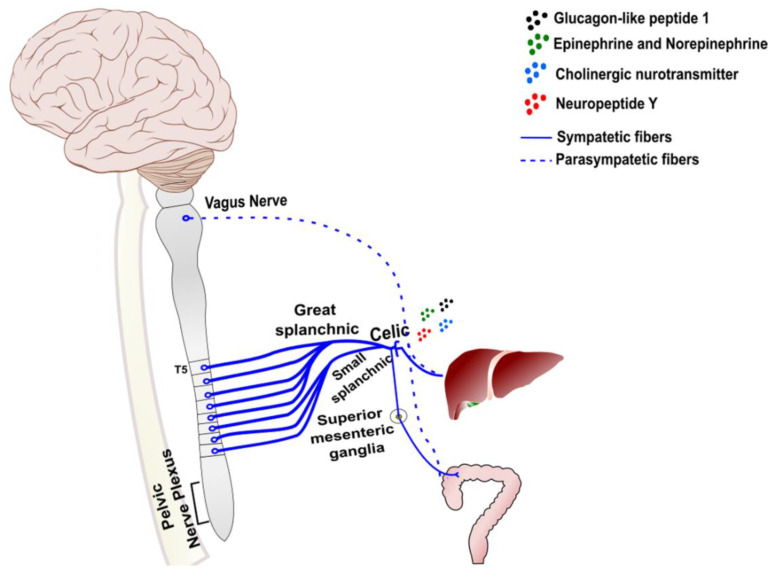
Arrangement of autonomic fibers in the liver. The interaction of the liver with the nervous system takes place through communication between the peripheral nerves and the vagus nerve, which largely innervates the peritoneal cavity. Sympathetic fibers that innervate liver tissue originate from neurons in the celiac and superior mesenteric ganglia. Furthermore, parasympathetic fibers originate mainly from preganglionic neurons located in the dorsal motor nucleus of the vagus nerve. Both are capable of interactions mediated by the release of adrenergic and cholinergic neurotransmitters and neuropeptides, such as neuropeptide Y, (continue on next page) leading to neuroendocrine-immune interactions that contribute to the modulation of inflammatory responses.

**Table 1 cells-11-02327-t001:** Clinical experiences of immunological tolerance in liver transplant recipients.

Number of Patients	Type of Graft	Immunosuppressive Therapy	Patient Selection	Time from Transplantation to Weaning	Complete IS Withdrawal	Mean Follow-Up Period after Withdrawal	Ref.
63	Living donor	Tacrolimus	Patients who survived more than 2 years after transplantation, maintained good graft function, and had no rejection episodes in the preceding 12 months	24 months	38.1%	23.5 months (range, 3–69 months)	[31]
45 (adults)	Cadaveric donor	Tacrolimus: 93%, Cyclosporin: 7%	>3 years after liver transplantation, >12 months without rejection, no autoimmune disease	43.2 months (mean; SD 0.96)	22.2% (no difference with the control group)	26 months (range 11–36)	[32]
34 (adults)	Cadaveric donor	Ciclosporin	>1 year after liver transplantation, positive for hepatitis C virus RNA, absence of rejection or cirrhosis on biopsy	63.5 months (mean; SD 20.1)	23.4%	45.5 months (mean; SD 5.8)	[33]
12 (adults)	Cadaveric donor	Ciclosporin	≥2 years after liver transplantation; ≥1 year with no rejection; no autoimmune disease, cancer, or viral disease	57.5 months (mean; SD 33.5)	38%	10–30 months (mean)	[34]
5 (children), median age: three years old (range, 8 months to 9 years)	Four parental living donors and one cadaveric donor	Tacrolimus	Patients who had a very low tacrolimus trough level (<1 ng/mL by liquid chromatography-mass spectrometry	45 months (range, 14 months to 60 months)	100%	32 months (range, 14 months to 82 months)	[35]
20 (children), <18 years old	Parental living donor	Tacrolimus: 65%, Ciclosporin: 35%	Allograft function while taking a single immunosuppressive drug, no evidence of acute or chronic rejection or significant fibrosis on liver biopsy	>48 months	60%	32.9 months (median; IQR 1.0–49.9)	[36]
102 (adults)	Not specified	Tacrolimus: 38.8%, Ciclosporin: 26.5%, Mycophenolate mofetil: 17.3%	Comorbidities of immunosuppression, risk of neoplasm	103 months (mean; SD 47)	40.2%	48.9 months (mean)	[37]
24 (adults)	Cadaveric donor	Tacrolimus: 20.8%, Ciclosporin: 8.3%, Mycophenolate mofetil: 29.2%, Sirolimus: 8.3%, Monotherapy: 66.7%	>3 years after liver transplantation, no active hepatitis C virus infection, no autoimmune disease	112 months (median; IQR 72–160)	62.5%	14 months (median)	[38]
34 (adults)	Not mentioned	Tacrolimus: 53%, Ciclosporin: 26.5%	Positive for hepatitis C virus RNA, identified as highly specific for operational tolerance	86 months (mean; SD 37)	50%	12 months (mean)	[39]
15 (adults), ≥18 years old	Cadaveric or living donor	Calcineurin inhibitor to sirolimus (SRL)	Adult LTR ≥ 18 years of age, ≥3 months of sirolimus monotherapy with trough levels of 3–8 ng/mL, ≥3 years post-LT (primary living or deceased donor)	6.7 ± 3 years	53%	12 months	[40]
88 (children), median age: 11 years old	Not mentioned	Tacrolimus	Alanine aminotransferase or gamma glutamyl transferase level exceeding 100 U/L, liver transplant recipient at ≤6 years of age, ≥4 years after transplant, no acute or chronic rejection within 2 years	36–48 weeks	37.5%	48 months	[36]

IS: Immunosuppressant; SD: Standard deviation; LT: Liver transplantation; LTR: Liver transplant recipient; IQR: Interquartile range.

**Table 2 cells-11-02327-t002:** Immune cells and the mechanism of tolerance after liver transplantation.

Immune Cells	Type of Study (Pre-Clinical or Clinical)	Mechanism	Type of Approach	Outcome	Reference
NK cells	Experimental—rats	Immunomodulatory effect mediated by NK cell activation through a receptor	Activation of NK by αGalCer receptor after OLT	NK cell activation by the αGalCer receptor was capable of inducing an anti-inflammatory profile, increasing IL-10 and decreasing IFN-γ	[44]
	Experimental—rats	IDO expressed on the NK cell surface mediating an immunomodulatory response	Induction of NK in an immunomodulatory microenvironment by IL-14 after OLT	Donor IL-4 injection induced the expression of IDO in NK cells and alternatively activated macrophages to increase the tolerance response after LT	[45]
	Experimental—rats	Enhancement of donor liver NK cells to prevent acute rejection	Donor NK liver cells infusion through portal vein (3 × 10^6^ cells) of recipients	Infusion of donor liver NK cells could downregulate the acute rejection microenvironment, but no induced spontaneous tolerance was observed after OLT	[46]
imDCs	Experimental—rats	Overexpression of IL-10, FasL, or TGF-β on DCs ameliorated liver damage after HLT	i.p. or i.v. injection of imDCs overexpressing IL-10 or FasL (2 × 10^6^)	Injection (i.p.) of imDCs overexpressing IL-10 or FasL prevented liver damage and probably induced Treg cells through the regulatory milieu	[47,48,49]
DCs	Clinical trial—NCT03164265	Infusion of DCs from a donor (phase I/II)	Donor DC cells were infused 7 days before the LT (2.5–10 × 10^6^/kg)	The trial is ongoing. Donor DCs were able to maintain a regulatory profile and suppress alloreactive cells against the donor cells	[50]
	Clinical trial— NCT04208919	Infusion of DCs from a donor (phase I/II)	Donor DC cells were infused 1 and 3 years after the LT (3.5–10 × 10^6^/kg)	The results have not been published yet but seem highly promising	clinicaltrials.gov identifier number: NCT04208919
Treg cells	Experimental—in vitro	Induction of Treg cells by exogenous IL-2 in the culture	IL-2 was added in culture of cells obtained from rats with tolerogenic, synergistic, and rejection groups after OLT	The addition of IL-2 in the culture was capable of inhibiting effector T cell differentiation and increasing the regulatory milieu in a dose-dependent manner	[51]
	Clinical trial—NCT02474199	Infusion of DARTreg (phase I/II)	DARTreg infusion intravenous 300–500 × 10^6^ cells/kg	The trial showed safety and the capacity to induce a tolerogenic profile	clinicaltrials.gov identifier number: NCT02474199
	Clinical trial— NCT03577431	Infusion of arTreg (phase I/II)	arTreg-CSB intravenous infusion 1–2.5 × 10^6^ cells/kg	The results have not been published yet but seem highly promising	clinicaltrials.gov identifier number: NCT03577431
	Clinical trial— NCT01624077	Infusion of DARTreg (phase I)	DARTreg infusion 1 × 10^6^ cells/kg	The trial showed safety and the capacity to induce a tolerogenic response	clinicaltrials.gov identifier number:NCT01624077
	Clinical trial— NCT02166177	Infusion of an autologous Treg product—polyclonal treg (phase I/II)	0.5–4.5 × 106 cells/kg	No result posted	clinicaltrials.gov identifier number:NCT02166177

DC: Dendritic cells; NK: Natural killer; Treg: Regulatory T; OLT: Orthotropic liver transplantation; HLT: Heterologous liver transplantation; DARTreg: Donor alloantigen-reactive Treg cells; arTreg: Alloantigen-reactive Treg cell.

**Table 3 cells-11-02327-t003:** Expression of adrenergic and cholinergic receptors in primary T lymphocytes. Only studies using primary cells that positively identified the expression of neurotransmitter receptors in T lymphocytes were considered.

Receptor		Cell Type	Form of Detection	Biological Effect	Reference
Adrenergic Receptors					
β1AR		Spleen LøT	qPCR IFI	ND	[122]
		LøT_CD4_	qPCR	Switching from a Th1 cytokine profile to a Th2 cytokine profile	[123]
β2AR		Th1 Cells	IFI Terbutaline stimulation	Inhibition of IFN-γ production Inhibition of IgG1 production by B cells	[124]
		Naïve LøT_CD4_ Th1 Cells	RT-PCR NE stimulation	Decrease in IL-2 production	[125]
		Spleen LøT	qPCR IFI	ND	[122]
		Naïve T Lø Treg Lø	WB WB, IFI	Increased suppression of naïve T Lø activation in vitro	[121]
		LøT_CD4_	qPCR	Switching from a Th1 cytokine profile to a Th2 cytokine profile	[123]
		Treg Lø	In silico analyses nCounter RNA analyses WB	ND	[126]
		Naïve LøT_CD8_ Activated LøT_CD8_	WB	Inhibition of naïve LøT_CD8_ activation	[127]
β3AR		Con A-stimulated TLø	RT-PCR	Inhibition of cytokine mRNA accumulation	[128]
		Spleen LøT	qPCR IFI	ND	[122]
		LøT_CD4_	qPCR	Switching from a Th1 cytokine profile to a Th2 cytokine profile	[123]
α2AAR		LøT_CD4_ LøT_CD8_ Treg Lø	In silico analyses nCounter RNA analysis WB	ND	[126]
Cholinergic Receptors					
m1		PBL (T/B cell enriched)	RT-PCR	Increased IL-2 production	[129]
		Spleen LøT_CD4_ and LøT_CD8_	qPCR	Th2 and Th17 responses	[130]
m2		PBL (T/B cell enriched)	RT-PCR	Increased IL-2 production	[129]
m3		LøT LøT_CD4_	RT-PCR	ND	[131]
		Spleen LøT_CD4_ and LøT_CD8_	qPCR	Th2 and Th17 responses	[130]
		LøT_CD4_	In silico analyses nCounter RNA analyses WB	ND	[126]
m4		LøT LøT_CD4_	RT-PCR	ND	[131]
		Spleen LøT_CD4_ and LøT_CD8_	qPCR	Th2 and Th17 responses	[130]
		Treg Lø ^2^ LøT_CD4_ ^2^ LøT_CD8_	In silico analyses nCounter RNA analyses WB	ND	[126]
m5		LøT LøT_CD4_	RT-PCR	ND	[131]
		Spleen LøT_CD4_ and LøT_CD8_	qPCR	ND	[130]
α	2	Spleen LøT_CD8_	qPCR	ND	[130]
	4	Activated LøT_CD4_ Activated LøT_CD8_	qPCR	ND	[130]
	5	Spleen LøT_CD4_ and LøT_CD8_	qPCR	Th1 polarization	[130]
	7	Treg Lø	RT-PCR FITC-labeled α-bungarotoxin	Increased CTLA-4 expression	[132]
		Activated LøT_CD4_ Activated LøT_CD8_	qPCR	ND	[130]
	9	Spleen LøT_CD4_ and LøT_CD8_	qPCR	Th1 polarization	[130]
	10	Spleen LøT_CD4_ and LøT_CD8_	qPCR	Th1 polarization	[130]
β	1	Spleen LøT_CD4_ and LøT_CD8_	qPCR	ND	[130]
		LøT_CD8_	In silico analyses nCounter RNA analyses WB	ND	[126]
	2	Spleen LøT_CD4_ and LøT_CD8_	qPCR	Th1 polarization	[130]
		Treg Lø LøT_CD4_ LøT_CD8_	In silico analyses nCounter RNA analyses WB	ND	[126]
	4	Spleen LøT_CD4_ and LøT_CD8_	qPCR	Th1 polarization	[130]

AR: Adrenergic Receptors; β2AR: β2 Adrenergic Receptors; ND: Not Determined; WB: Western blot; IFI: Indirect Immunofluorescence; NE: Norepinephrine; PBL: Peripheral Blood Leukocytes; qPCR: Quantitative RT-PCR; ^2^ Memory phenotype.

## Data Availability

Not applicable.

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
