# Peer review of "Immunological Tolerance in Liver Transplant Recipients: Putative Involvement of Neuroendocrine-Immune Interactions"

_cells, 2022, doi:10.3390/cells11152327_

Round 1

Reviewer 1 Report

Authors reported a comprehensive review on immunological tolerance in liver transplant recipients, especially regarding neuroendocrine-immune interactions.

They described their putative mechanism through PAMPS and MAMPS from commensal flora.

The review is well written and thoroughly investigated.

I have several comments for the authors.

1.     The relationship between gut and liver and brain interaction and their effect on T reg induction and maturation should be hot topic. Authors should comment on those articles.

The liver-brain-gut neural arc maintains the Treg cell niche in the gut.

Teratani T, et al. Nature. 2020;585(7826):591-596.

2.     Most recent related articles should be referred, eg. Clinical liver transplant tolerance: Recent topics.Takatsuki M, et al. J Hepatobiliary Pancreat Sci. 2022;29(3):369-376.

Reviewer 2 Report

Jaciara Fernanda Gomes Gama et al have written a very nice Review. It is well structured and the most important aspects are clearly presented. One small addition would be worth considering. In the context of inflammation, the resolution of inflammation plays at least as important a role. Therefore, the article would gain in impact if a few sentences could be written about this
